# Extremely large magnetoresistance in twisted intertwined graphene spirals

Yiwen Zhang[1,2,4], Bo Xie [1,2,4], Yue Yang[1,4], Yueshen Wu[1,2,4], Xin Lu [1], Yuxiong Hu[1], Yifan Ding[1,2], Jiadian He[1,2], Peng Dong[1,2], Jinghui Wang[1,2], Xiang Zhou [1,2], Jianpeng Liu [1,2,3] ✉, Zhu-Jun Wang [1] ✉ & Jun Li [1,2] ✉

Extremely large magnetoresistance (XMR) is highly applicable in spintronic devices such as magnetic sensors, magnetic memory, and hard drives. Typically, XMR is found in Weyl semimetals characterized by perfect electron–hole symmetry or exceptionally high electric conductivity and mobility. Our study explores this phenomenon in a recently developed graphene moiré system, which demonstrates XMR owing to its topological structure and high-quality crystal formation. We investigate the electronic properties of three-dimensional intertwined twisted graphene spirals (TGS), manipulating the screw dislocation axis to achieve a rotation angle of 7.3°. Notably, at 14 T and 2 K, the magnetoresistance of these structures reaches $1.7 \times 10^7$%, accompanied by a metal–insulator transition as the temperature increases. This transition becomes noticeable when the magnetic field exceeds a minimal threshold of approximately 0.1 T. These observations suggest the possible existence of complex, correlated states within the partially filled three-dimensional Landau levels of the 3D TGS system. Our findings open up possibilities for achieving XMR by engineering the topological structure of 2D layered moiré systems.

Giant magnetoresistance (GMR), typically below 100%, is a common phenomenon in some metallic thin films and manganese-based perovskites and plays a crucial role in spintronic devices[1,2]. The robust response to a weak external field has long been regarded as a valuable functional characteristic in industrial technology, representing a central objective for physicists and materials scientists alike. Conventional materials with large magnetoresistance (MR) are primarily manganites, exhibiting colossal magnetoresistance (CMR)[3–6] values up to $10^5$%, which is generally due to the alignment of the spin configuration under an external field. With the emergence of two-dimensional materials, some non-magnetic materials demonstrate extremely large magnetoresistance (XMR) ($10^3$%–$10^8$%), characterized by a substantial increase in resistivity under magnetic fields, even without saturation at very high fields[7–13], indicating versatile potential applications. This XMR, fundamentally distinct from GMR or CMR, is often observed in semimetals with topological properties and balanced electron-hole carriers, as collected in Fig. 1b[7].

Notably, van der Waals semimetals like $WTe_2$[8], $MoTe_2$[9,10], and $NbSb_2$[11] display XMR ($1 \times 10^5$% at 2 K). Angle-resolved photoemission spectroscopy (ARPES) studies in these materials reveal electron and hole Fermi surfaces of equal size[12], indicating a resonant compensated semimetal nature. Similarly, the 3D transition metal diphosphides, $WP_2$ and $MoP_2$, showcase promising XMR due to closely neighboring Weyl points. Other semimetals like $LaSb$[13], $TaAs$[14,15], $NbP$[16], $Cd_3As_2$[17], and $PtSn_4$[18] also exhibit XMR, mainly due to nearly perfect electron-hole compensation or high conductivity and mobility.

Graphite, a fundamental semimetal, has shown impressive XMR progress, attributed to its topological defects such as stacking faults and sample quality[19]. In twisted graphene/boron-nitride (BN) heterostructure[20], magnetoresistance (MR) values reach 880% at 400 K (9 T). An XMR of 110% at 300 K (0.1 T) was reported in twisted BN/

[1]School of Physical Science and Technology, ShanghaiTech University, Shanghai, China. [2]ShanghaiTech Laboratory for Topological Physics, ShanghaiTech University, Shanghai, China. [3]Liaoning Academy of Materials, Shenyang, China. [4]These authors contributed equally: Yiwen Zhang, Bo Xie, Yue Yang, Yueshen Wu. ✉e-mail: liujp@shanghaitech.edu.cn; wangzhj3@shanghaitech.edu.cn; lijun3@shanghaitech.edu.cn

**a**

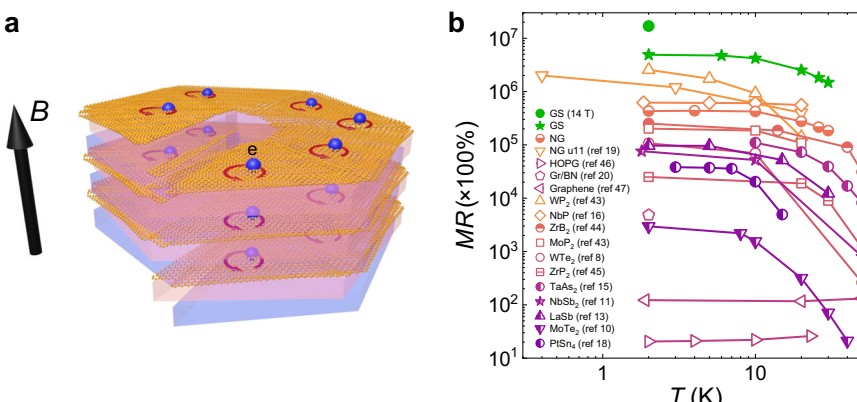

**Fig. 1 | XMR in various semimetals at an applied field of 7 T. a** The blue sphere represents the electron, and the red arc represents the partial trajectory of the electrons in GS. **a** Illustration of charge transport in GS. **b** The temperature dependence of larger MR for GS and some other typical materials, including the topological semimetals WP$_2$[43], NbP[16], MoP$_2$[43], WTe$_2$[8], ZrP$_2$[45], ZrB$_2$[44], TaAs$_2$[15], NbSb$_2$[11], LaSb[13], MoTe$_2$[10], and PtSn$_4$[18], highly oriented pyrolytic graphite (HOPG)[46], natural graphite (NG)[19], graphene[47], and graphene/BN moiré lattice[20].

graphene/BN, linked to Dirac plasma in the moiré system[21]. A record MR value of $1 \times 10^7\%$ (21 T) in micrometer-sized natural graphite suggests the importance of moiré patterns in XMR.

The 2018 discovery of superconductivity in magic-angle twisted bilayer graphene marked a milestone[22,23], making moiré superlattices a key focus in condensed matter physics[24–28]. These layers exhibit unique phenomena due to the interplay between graphene's Dirac wavefunctions and the interlayer moiré potential[29–32], leading to flat bands with nontrivial topology[25,33–40]. Recently, a three-dimensional bulk moiré graphite superlattice, consisting of alternating twisted graphene layers, was synthesized[41]. This structure, theoretically capable of 'magic momenta' and 3D Landau levels[42], maybe a platform for studying correlated and topological physics.

Our study explores the transport properties of this bulk moiré graphite, comprising hundreds and thousands of alternating twisted graphene multilayers stacked along a spiral dislocation, as schematically shown in Fig. 1(a). The twisted graphene spirals (TGS) exhibit extremely high carrier mobility ($\mu_m = 3 \times 10^6$ cm$^2$V$^{-1}$s$^{-1}$), forming Landau levels under low magnetic fields (~ 0.1 T). We observed an XMR of $1.7 \times 10^7\%$ at 2 K under a 14 T magnetic field in this 3D TGS system.

## Results and discussion
### Extremely large magnetoresistance
In Fig. 1b we present the temperature-dependent MRs (at $B = 7$ T) of several typical high-MR materials including Weyl semimetals[8–10,12–14,43], Dirac semimetals[11,17,18,44], nodal-line semimetals[45] (according to the degeneracy and momentum space distribution of the nodal points), natural graphite (NG)[19] and highly oriented pyrolytic graphite (HOPG)[46], graphene[20,47], and twisted spiral graphene presented in this work. Clearly, twisted spiral graphene holds the record-high MR in this moderate magnetic field regime. The MR at $T = 2$ K and $B = 7$ T is up to about $5 \times 10^6\%$, which is a record high value among all MRs measured at moderate magnetic fields (<~ 10 T) reported in the literature. The experimental setup for resistivity measurement is illustrated in Supplementary Fig. 1. The value of MR in another sample reaches $1.7 \times 10^7\%$ with applied field $B = 14$ T at $T = 2$ K.

For large MR, various potential mechanisms have been proposed, including electron-hole compensation[12,48], steep band[49], ultrahigh mobility[50], high residual resistivity ratio (RRR)[51,52], topological fermions[16,53], etc[54]. Mechanisms mentioned above can be analyzed through quantum oscillation and Hall effect measurement. Furthermore, a large MR is correlated with other systems, such as 2D materials, in which carriers are limited to the 2D plane, and the electronic structure largely depends on the thickness of the material. However,

the MR of the HOPG reveals MR of only about $1 \times 10^1\%$[19,46,47,55], being several orders of magnitudes smaller than that of the bulk twisted GS. Nevertheless, an extremely high MR of about $1 \times 10^6\%$ (2 K, 7 T) has also been observed in the Sri Lankan NG, for which the existence of highly conducting 2D interfaces aligned parallel to the graphene planes is attributed to the origin[19,46,47,55].

Another system concentrated here is topological materials, which are characterized by symmetry-protected band crossing at or near the Fermi level in the Brillouin zone. Particularly, the present record of the maximum MR was reported in type-II Weyl semimetal WP$_2$ ($2 \times 10^6\%$, 7 T, 2 K)[43]. Such spectacularly large MR in WP$_2$ were attributed to both the nontrivial wave function textures around Weyl nodes and the presence of compensating electron and hole pockets in the Fermi surface, where the former can suppress back scatterings and induce considerably large mobilities[16,17,43,56]. As reported in NG[19], the large MR in graphite shares a different origin from Weyl semimetals. Due to the advanced growth method proliferating the scope of graphene moiré superlattices[41], bulk graphite comprises alternatingly twisted graphene multilayers, which are sequentially stacked along a spiral dislocation. Notably, such spiral structure is thermodynamically stable[57], and some more detailed information on GS growth can be seen in Supplementary Information. However, the topological structure of GS still deserves discussion, and the structure of the GS will be introduced in the following section.

### Structure of the twisted graphene spiral
In scanning electron microscopy (SEM) imaging, we observe stepwise variations in contrast, allowing us to discern up to ten individual graphene layers[58]. Given that each graphene spiral (GS) in the crystal is comprised of several dozen turns, SEM thus is not adept at distinguishing the contours of GS with up to ten layers. To circumvent this limitation, we resort to confocal laser scanning microscopy (CLSM), which possesses the ability to simultaneously generate intensity and topographical images. This facilitates the measurement of the in-plane dimensions and the out-of-plane stacking sequence of the as-grown GSs[59]. Through CLSM, we confirm that the chemical vapor deposition (CVD) approach is proficient at producing GSs with dimensions spanning hundreds of micrometers, revealing interlayers characterized by an approximate twist angle of 7° (as showcased in Fig. 2a and Supplementary Fig. 9). In addition, the adjacent zigzag edges of these spirals tilt at an angle of 7°, a detail captured in the atomic force microscopy (AFM) imagery presented in Supplementary Fig. 9. Such evidence intimates that the freshly synthesized GSs adopt an intertwined helical configuration, characterized by a twist of 7°.

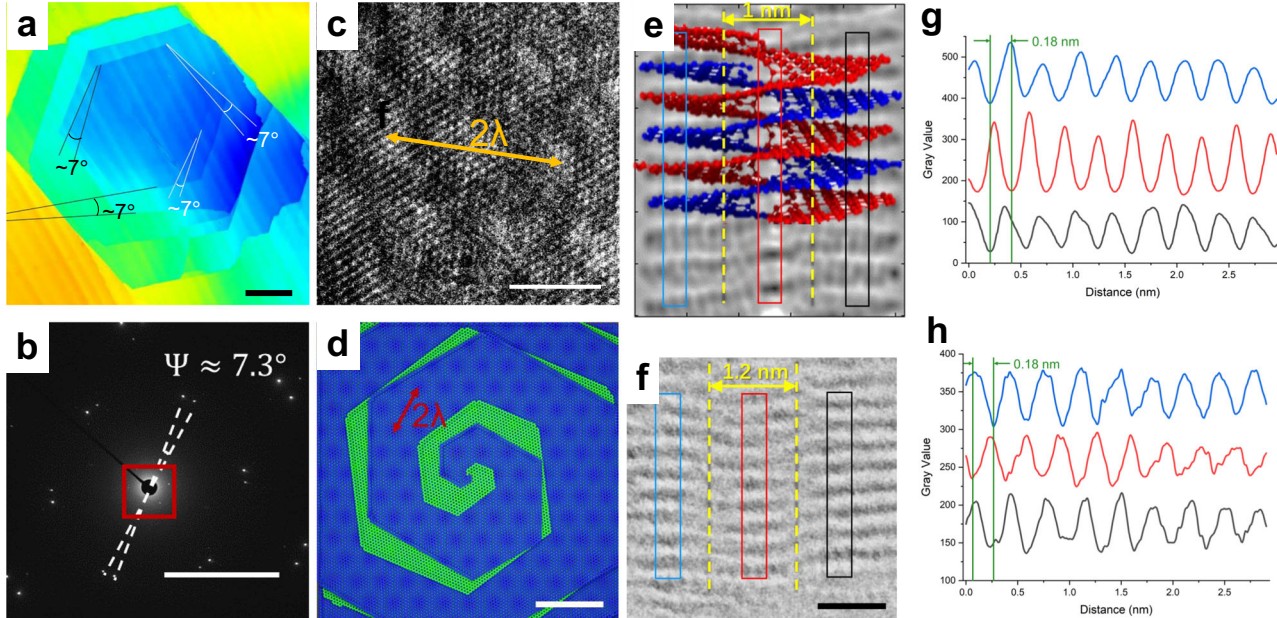

**Fig. 2 | Structure analysis of the twisted graphene spiral. a** The CLSM image provides height information, illustrating the central region of an individual GS. In the image, blue denotes lower heights; scale bar, 50 μm. **b** confirming an average twist angle of $\Psi \approx 7.3°$, which is consistent with the measurements obtained from AFM (shown in Supplementary Fig. 9). The red rectangle in (**b**) highlights the presence of diffraction points corresponding to the moiré pattern, as further depicted in Supplementary Fig. 9; scale bar, 2 1/nm. **c** The moiré pattern is a periodic arrangement of bright (AA stacking area) and dark (AB stacking area) regions; scale bar, 2 nm. **d** Schematic diagram illustrating the structure of the GS; scale bar, 5 nm. **e** Side view of the double helix GS model and the corresponding simulated STEM image. **g** Contrast line profile along the perpendicular plane across the colored strips in (**e**). A staggered series of graphene planes are signatures of double helical organization, as its simulated STEM images show. **f** Magnified STEM image of GS showing double helix feature. **h** The contrast line profile analysis of (**f**) exhibits similar features as shown in the simulated images (**e**, **g**). TEM operating voltage was 80 kV in order to minimize beam damage; scale bar, 1 nm.

To further validate the presence of an intertwined helical structure with a homogeneous twist angle, we transferred the identical GS onto a transmission electron microscopy (TEM) grid for TEM diffraction measurements, following the transport measurements (which will be discussed later). The TEM observations, along with the corresponding selected area electron diffraction (SAED) patterns, reveal that the twist angle of the GS is 7.3 ± 0.176° (Fig. 2b and Supplementary Fig. 9). Significantly, the twist angle determined through SAED is consistent with the findings from CLSM and AFM observations in Fig. 2a and Supplementary Fig. 9. More accurate twist angle measurements were made by diffraction mode of TEM, which allows for the observation of superlattices formed by a pair of graphene lattices in the diffraction pattern, as depicted in Fig. 2b and Supplementary Fig. 9. It is worth mentioning that electron diffraction of moiré patterns can also be observed, exhibiting the same symmetry as the original lattice.

The relative rotation of the GS leads to the formation of a periodic moiré superlattice within the helical graphene structure. The wavelength of this superlattice ($\lambda$) is determined by the twisted angle ($\theta$) present in the moiré superlattice, $\lambda = a/(2\sin(\theta/2))$ ($a$: the lattice constant of the monolayer graphene)[60]. High-resolution TEM characterization reveals the presence of a moiré pattern with a twist angle of $\theta = 7.3°$, corresponding to a superlattice wavelength of ~ 1.93 nm. Figure 2c clearly shows the moiré pattern as a periodic arrangement of bright and dark regions, with a wavelength of ~1.93 nm, confirming that it corresponds to the twist angle of 7.3°. Based on these comprehensive characterizations, we conclude that the as-grown GSs consist of intertwined helical structures with a 7° in-plane twist, as depicted in the schematic illustrations in Fig. 2d.

To obtain a precise characterization of the out-of-plane structure of the GS and further confirm the presence of intertwined helical structures, cross-sectional TEM samples were prepared. These samples were extracted from the identical GS after performing CLSM observations (Supplementary Fig. 7), TEM measurements (Fig. 2b), and magnetoresistance measurements (Figs. 1, 3). The site-specific focused ion beam (FIB) lift-out technique was employed to prepare the GS cross-sectional samples. The process of FIB preparation for observing the cross-sectional structure of the GS is depicted in Supplementary Fig. 7.

The cross-sectional scanning transmission electron microscopy (STEM) observations (Supplementary Fig. 8) reveal a double staggered series of graphene planes, suggesting the presence of a screw dislocation with a double helical structure. In order to gain a better understanding of the cross-sectional structure of the screw dislocation in the graphene multilayers, we constructed a structural model for the double helical structure, which is further relaxed using molecular dynamics simulations. Then, we perform STEM intensity simulations based on the fully relaxed double helical structure using QSTEM code package[61]. Figure 2e presents a schematic model of the double helical structure of GS, accompanied by simulated STEM images derived from the fully relaxed structural model. Figure 2e–h presents a comparison between the configuration of the simulated double helical structure and the experimentally observed images (Fig. 2e, f). While the atomic structure cannot be completely determined from the cross-sectional STEM image, the stacking configuration and curvature closely resemble the expected double helical structure. This can be visualized as a result of cutting through the $sp^2$-bonded graphene layers perpendicularly, causing them to slip along the cutting line by two layers and reconnecting the bonds. Line profiles were plotted perpendicular to the lateral graphene layers, highlighting the distinctive configuration with double staggered graphene planes (Fig. 2g, h). This configuration involves a helical path traced around the linear defect (spiral core) as the atomic planes slip within the crystalline lattice. Indeed, a comparison of the line profile between the simulated double helical

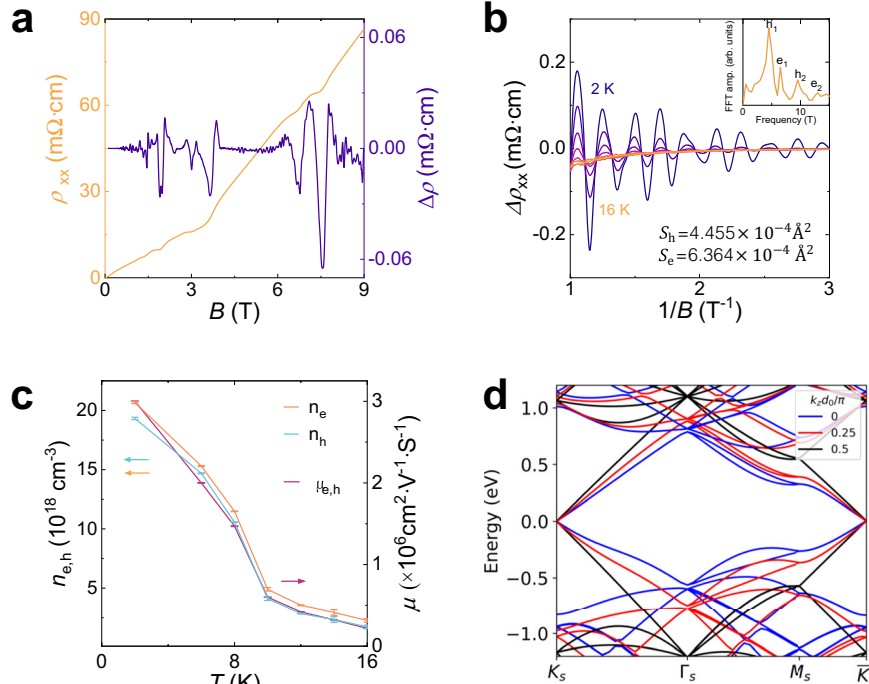

**Fig. 3 | Quantum oscillation. a** Left axis: Resistivity $\rho_{xx}$ versus $B$ measured at $T = 2$ K for GS. Right axis: Background removed data $\Delta\rho_{xx}$ showing quantum oscillations measured over different magnetic field regions. **b** Corresponding $\Delta\rho_{xx}$ data at different temperatures from 2 K to 16 K. The inset shows an FFT result at 2 K. Peaks $h_{1,2}$, $e_{1,2}$ correspond to the 1st and 2nd harmonics of oscillations from holes and electrons. $S_h$ and $S_e$ indicate the Fermi surface area of holes and electrons. **c** Temperature dependence of the carrier concentration (left ordinate) and the

mobility (right ordinate). A relatively high carrier concentration and mobility can be observed at 2 K. The error bar indicates the 95% confidence interval. **d** The electronic band structures of GS in the $(k_x, k_y)$ plane. The blue lines represent the electronic band structures for $k_z d_0/\pi = 0$. The red lines represent the electronic band structures for $k_z d_0/\pi = 0.25$. The black lines represent the electronic band structures for $k_z d_0/\pi = 0.5$.

structure and the experimental observation reveals a remarkable similarity. This similarity is characterized by an integral-sliding pattern within the spiral core, where the atomic planes exhibit a shift equivalent to half a lattice of $sp^2$ carbon layers. The line profile analysis clearly shows a shift in the layered stacking configuration of approximately 0.18 nm (half the interlayer spacing of graphite) within the spiral core region (highlighted by the red stripes in Fig. 2e, f), compared to the surrounding areas (blue and black stripes). The remarkable consistency between the experimental line profile of the cross-sectional GS and the simulated image further confirms the proposed double helical structure.

## Quantum oscillation

As well studied in previous works, a twist angle between two adjacent graphene layers can fundamentally change their electronic band structures, leading to topological flat bands which are responsible for various novel phenomena in electronic transport properties. The transport properties of the bulk twisted GS were investigated in conventional four-terminal methods, and magnetic fields were applied parallel to the c-axis. The NG was also studied as a comparison. A typical low-temperature data for in-plane longitudinal resistivity ($\rho$) of twisted GS as a function of the magnetic field from $B = 0 - 9$ T is shown in Fig. 3a. $\rho$ increases colossally with the magnetic field, and, at 9 T, it is almost 7 orders of magnitude larger than the zero-field value, which will be analyzed in next section. In addition, Shubnikov–de Haas (SdH) oscillations are superimposed on the large magnetoresistance background. These oscillations, which start at a low magnetic field $B \approx 0.1$ T, can be better observed in the background-removed data plotted in Fig. 3a and b. The final quantum oscillation appears at 4.67 T (indicating a hole) and at 6.65 T (indicating an electron). The background can be removed by subtracting a

smoothed data curve. Figure 3b shows oscillations under a smaller magnetic field region at different temperatures from 2 K to 16 K. Two series of oscillations can be distinguished. In the Fourier transformation of the $\Delta\rho$ versus $(1/B)$ data shown in Fig. 3b inset, two frequencies are found and assigned to the hole pocket (4.67 T) and electron pocket (6.67 T). Peaks $h_{1,2}$, $e_{1,2}$ correspond to the 1st and 2nd harmonics of oscillations from holes and electrons. Thus, the area of the Fermi surface ($S_h = 4.455 \times 10^{-4}$ Å$^2$ for holes and $S_e = 6.364 \times 10^{-4}$ Å$^2$ for electrons) can be obtained through the function $F = (\hbar/2\pi e)A_F$, where $F$ is the FFT frequency, $\hbar$ is reduced Planck's constant, $e$ is the elementary charge, $A_F$ is the area of Fermi surface. Carrier mobility and concentration can reflect quasi-particle properties near the Fermi level, which are two important parameters of a material that can be derived from the Hall coefficient.

Hall effect measurements have been performed in field sweep mode. The field dependence of the Hall resistivity ($\rho_{xy}$) exhibits a nonlinear behavior in low fields, indicating the involvement of more than one type of charge carrier in the transport properties. The nonlinear Hall curve maybe described by the two-carrier model[62]:

$$\rho_{xy} = \frac{1}{e} \frac{(n_h\mu_h^2 - n_e\mu_e^2) + \mu_h^2\mu_e^2 B^2(n_h - n_e)}{(n_h\mu_h + n_e\mu_e)^2 + \mu_h^2\mu_e^2 B^2(n_h - n_e)^2} B \qquad (1)$$

where $n_e(n_h)$ and $\mu_e(\mu_h)$ are the carrier density and mobility of electrons (holes), respectively. The mobilities of electron-type and hole-type are assumed to be equal at low temperatures, as both of them are quasi-particle excitations around the Dirac points, which have similar scattering mechanisms. The fitting result can be seen in Fig. 2c. The electron (hole) carrier concentration is found to be $n_e = 2.07 \times 10^{19}$ cm$^{-3}$ ($n_h = 1.93 \times 10^{19}$ cm$^{-3}$) at 2 K, with a net electron carrier concentration $\delta n_e = 1.4 \times 10^{18}$ cm$^{-3}$. The corresponding mobility exhibits a high value

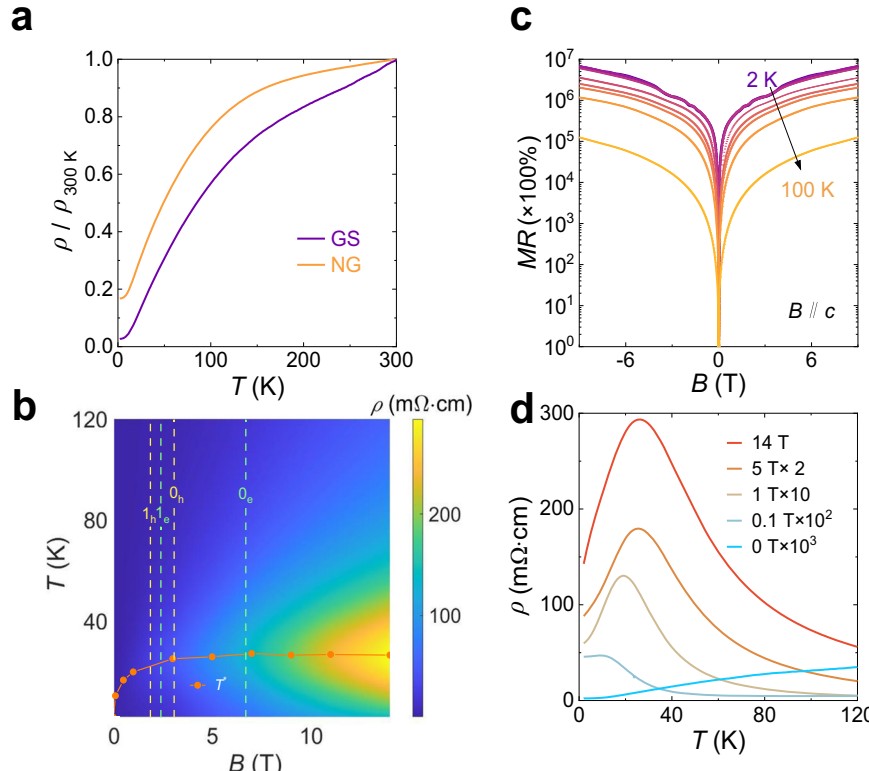

**Fig. 4 | Anomalous metal−insulator transition in twisted graphene spiral. a** The normalized temperature dependent resistivity of GS and NG under zero field. **b** A color map of resistivity in the plane of the temperature and magnetic field. The orange scatterline indicates the magnetic field dependence of the anomalous transition temperature from metal to insulator. The vertical green (yellow) dashed lines mark the filling factors of Landau levels from the electron (hole) carriers, which are labeled as $0_e$ ($0_h$) and $1_e$ ($1_h$). **c** MR of GS at different temperatures from 2 to 100 K. The highest MR of $7 \times 10^6$% is observed at 2 K under 9 T. **d** Line cuts of resistivity versus temperature for a range of magnetic fields (shown are traces taken from 0 T to 14 T). An unconventional metal-insulator transition appears when a finite external magnetic field has been applied.

$\mu_{(e,h)} = 3 \times 10^6$ cm$^2$ V$^{-1}$ s$^{-1}$. This value is close to that of the Weyl semimetal NbP ($5 \times 10^6$ cm$^2$ V$^{-1}$ s$^{-1}$)[16]. The quantum oscillation behavior maybe qualitatively understood from the low-energy band structures of the bulk twisted intertwined GS. Away from the central dislocation core, the system can be viewed as numerous layers of TBG, which are coupled via nearest neighbor interlayer hopping. As a result, such bulk moiré superlattice is bestowed an additional vertical wavevector ($k_z$) degree of freedom. When the twist angle is relatively large ($\geq 3°$), the low-energy band structure consists of Dirac cones in the $k_x$-$k_y$ plane, with $k_z$-renormalized Fermi velocities, as shown in Fig. 2d. Neglecting the frame-rotation effects of the two graphene layers, it turns out that the low-energy states of the system have an approximate particle-hole symmetry at each vertical wavevector $k_z$. Consequently, the Dirac point at each $k_z$ is pinned to the same energy by virtue of this particle-hole symmetry. Including the frame-rotation effects would shift the energy of the Dirac point, leading to a $k_z$ dispersion of the form $4\gamma_0 cos^2 k_z d_0$[63], with $\gamma_0 \sim 3.5$ meV, leading to a $k_z$ dispersion of Dirac points $\sim 7$ meV. This may lead to the co-existence of electron and hole carriers even when the system is charge-neutral. Moreover, other effects, such as the variation of the interlayer distance (from 3.35 angstrom to 3.50 angstrom according to the TEM measurements) and remote-band renormalization effects may further enhance the $k_z$ dispersion of the Dirac points, which qualitatively explains the co-existing electron- and hole-type carriers in the system. As a two-band system with similar carrier concentration between electrons and holes, the compensation mechanism can be a major factor. Considering the effect of topological Landau levels on MR, we measured $R(T)$ curves under different magnetic fields.

## Anomalous metal−insulator transition

Figure 4 shows the temperature dependence of normalized $\rho$ of twisted GS and NG under zero magnetic field. The twisted GS shows a much larger RRR [$\rho$(300 K)/$\rho$(2 K)] = 38, which is considerably larger than that of NG (RRR = 6). Generally, the less residual resistivity demonstrates the high-quality of single crystals with less impurity scattering. However, the special lattice structure of the bulk twisted GS may also induce significant resistivity difference due to the dramatic changes in the band structures, as has been observed in twisted bilayer graphene[64,65].

A color map of resistivity in the plane of the temperature from 2 to 120 K and magnetic field from 0 to 14 T has been shown in Fig. 4b. An unconventional transition from metallic behavior (where resistivity $\rho$ increases with temperature) to insulator-like behavior (where resistivity $\rho$ decreases with temperature) appears when a finite external magnetic field has been applied. Transition temperature $T^*$ has been defined where d$\rho$/d$T$ = 0. The orange scatterline shows the field dependence of $T^*(B)$. As the magnetic field increases, the $T^*$ rises rapidly and finally saturates to ~ 26.9 K when the field is larger than 3 T. This is exactly the same magnetic field value above which the zeroth landau level starts to be filled, as marked by the rightmost yellow dashed line in Fig. 4b. Line cuts of resistivity versus temperature for several typical values of magnetic fields varying from 0 to 14 T can be seen in Fig. 4d. A monotonic temperature dependence where $\rho$ increases with temperature has been observed under zero magnetic field; and a nonzero $T^* \sim 2$ K emerges when the field is increased to 0.1 T, which is the same characteristic field amplitude for the onset of Landau level quantization. This implies that the anomalous metal-

insulator transition at temperature $T^*$ maybe attributed to the correlation effects of the electrons in partially occupied Landau levels. Especially in such a bulk twisted GS system, the linear Dirac dispersions and the quasi-2D Fermi surface would lead to weakly dispersive higher Landau levels and even dispersionless zeroth Landau levels in such a 3D bulk system, which may significantly boost electron-electron interaction effects.

Line profiles of MRs under different temperatures are presented in Fig. 4c. This XMR is concomitant with an anomalous metal-insulator transition with increasing temperature under magnetic fields, which is reminiscent of the isospin Pomeranchuck effect observed in magic-angle twisted bilayer graphene[66,67]. This phenomenon represents a modification of hadronic interactions arising from the increased nuclear spin entropy found in the solid phase, where atoms are spatially confined. Strongly correlated electron states manifest in magic angle twisted bilayer graphene (MTBG) owing to the presence of flat bands. However, in our system, the twist angle is significantly larger than the magic angle, thus, it is necessary to apply a finite magnetic field for the emergence of strongly correlated electron states. A temperature dependence of MR under a magnetic field ranging from 3 T to 14 T (shown in Supplementary Fig. 4) exhibits an ultrahigh MR reaching $3.4 \times 10^4$% under 14 T even at room temperature 300 K. Undisputedly, the topological dispersionless Landau level in such large-angle-twisted GS have enhanced the MR under a high magnetic field, although the specific mechanism still deserves further research.

In summary, our work has revealed an XMR and a remarkable metal-insulator transition in a bulk graphite superlattice arranged in a 3D TGS system. Through quantum oscillation measurements, we identified two distinct frequencies that correspond to separate charge carrier pockets: an electron pocket at 4.67 T and a hole pocket at 6.67 T. These findings are corroborated by Hall effect measurements. The simultaneous presence of both electron-type and hole-type carriers suggests a disruption in the particle-hole symmetries, possibly influenced by variations in the interlayer spacing and remote-band renormalization effects.

Upon applying an external magnetic field, we observed a notable transition from metallic to insulator-like behavior. The critical temperature, $T^*$, stabilizes around 26.9 K in fields above 3 T, indicating the onset of filling in the 3D zeroth Landau level. This behavior implies that the transition may stem from correlation effects among electrons in these partially filled 3D Landau levels. Accompanying this transition is an XMR, measured at approximately $1.7 \times 10^7$% at 2 K and 14 T, setting a record. This significant increase in magnetoresistance is linked with the metal-insulator transition and is reminiscent of the isospin Pomeranchuck effect observed in magic-angle twisted bilayer graphene.

Our results establish the TGS as a potential platform for studying XMR in 2D layered materials alongside correlated and topological phenomena typically observed in moiré graphite superlattices.

## Methods
### Structure analysis
Transmission electron microscopy analysis. The Polymethyl Methacrylate (PMMA) solution was spin-coated onto the GS crystal at a speed of 600 rounds per minute (rpm) for 5 s, followed by spinning at a speed of 3000 rpm for 60 s and then drying at room temperature for 10 min. Following the drying process, the prepared sample was transferred onto a gold grid, and the PMMA protective layer was removed through a soaking procedure in acetone. Subsequently, Aberration-corrected transmission electron microscopy was carried out using a JEOL Grand Arm 300F, operated at 80 kV to minimize the risk of inducing knock-on damage to graphene. Image acquisition was performed with an exposure time of 0.5 s on a OneView camera binned to $2\,k \times 2\,k$ pixel resolution. Post-processing of acquired images was undertaken for further analysis and enhancement of data quality.

### Transport measurements
The electrical transport measurements were performed in a physical property measurement system (PPMS-Dynacool, Quantum Design) using an external electric meter (Keithley 2400 as a current source and Keithley 2182a as a voltage meter). The transport properties of bulk twisted GS were investigated using conventional four-terminal methods.

### Lattice relaxation calculation
The structural relaxation is calculated utilizing Large-scale Atomic-Molecular Massively Parallel Simulation (LAMMPS)[68]. The interlayer interactions between adjacent layers are described using the Dispersion Interaction Random Phase (DIRP) potential, while the intralayer potentials are described by the Adaptive Intermolecular Reactive Bond Order (AIREBO) potential[69] with a cutoff of 3Å. These potentials have been extensively adopted in the molecular dynamics study of twisted bilayer graphene with large twist angles. As mentioned above, since our sample is much larger than the typical length scale of the dislocation core, the influence of the dislocation line on both the lattice structure and the electronic properties has been neglected. In our molecular dynamics simulations, we employ a moiré supercell with periodic boundary conditions applied in both in-plane and out-of-plane directions as the initial lattice structure. The twist angles between two layers, in accordance with the commensurate condition, are ± 7.34°. The corresponding lattice constant of the supercell is 1.92 nm. In order to find out the minima of the energy function, we utilize the steepest descent algorithm without considering the thermal effects. The convergence criterion for energy is set to $10^{-8}$eV. The settings in QSTEM included a voltage of 300 kV with a Cs value of 1.0 mm, which was crucial for accurate phase contrast simulations. The inner and outer angles are 2.06° and 8.31°, which are consistent with the actual experimental parameters. This integration aims to bridge the gap between mechanical stability and electronic properties. Through a comparison of the QSTEM and the STEM images, we have justified the reliability of the fully relaxed lattice structure. Then, we can investigate the electronic properties, including the electronic band structure, Fermi surface, surface states, and so on, utilizing the tight-binding model.

### Electronic band structure calculation
The calculation of the band structure of twisted GS is performed utilizing the tight-binding (TB) model[70,71] based on the fully relaxed structure. This model was first proposed by Moon and Koshino and is widely adopted in the twisted graphene community. The reliability of this model has been justified by directly comparing its band structures with the density functional theory calculations for twisted bilayer graphene[72,73]. To be specific, The Hamiltonian is written as

$$H = -\sum_{\{i,j\}} t\left(R_i - R_j\right)|R_i\rangle\langle R_j| + H.c. \tag{2}$$

where the hopping amplitude between two $p_z$ orbitals at different sites is expressed as

$$-t_{\mathbf{d}} = V_\sigma \left(\frac{\mathbf{d} \cdot \hat{\mathbf{z}}}{|d|}\right) + V_\pi \left[1 - \left(\frac{\mathbf{d} \cdot \hat{\mathbf{z}}}{|d|}\right)^2\right] \tag{3}$$

where $V_\sigma = V_\sigma^0 e^{-(r-d_c)/\delta_0}$ and $V_\pi = V_\pi^0 e^{-(r-a_0)/\delta_0}$ $d = (d_x, d_y, d_z)$ is the displacement vector between two sites. $d_c = 3.44$ Å is the interlayer distance. $a_0 = a/\sqrt{3} = 1.42$ Å is the distance between the two nearest neighbor carbon atoms. $V_\sigma^0 = 0.48$ eV is the transfer integral between vertically located atoms on the neighboring layers and $V_\pi^0 = -2.7$ eV is

that between the intralayer nearest-neighbor atoms. We set $\delta_0 = 0.184a$ so that the next-nearest intralayer coupling becomes $0.1\,V_\pi^0$.

## Data availability
The relevant data generated in this study are provided in the Supplementary Information.

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

## Acknowledgements

This research was supported in part by the Ministry of Science and Technology (MOST) of China (Nos. 2022YFA1603903 and 2020YFA0309601) (J.Li and J.Liu), the National Natural Science Foundation of China (Grants Nos. 12174257, 12004251, 12104302, 12104303, 12304217) (X.Z., J.W., and Y.W.), the Science and Technology Commission of Shanghai Municipality (Grant No. 21JC1405100) (J.Li), the Shanghai Sailing Program (Grant No. 21YF1429200) (J.W.), the start-up funding from ShanghaiTech University (J.Li). Thanks to the technical support from the Soft Matter Nanofab (SMN180827) and Center for High-resolution Electron Microscopy (C$\hbar$EM), SPST, ShanghaiTech University (EM02161943).

## Author contributions

Y.Z., J.Li, and Y.W. conceived and designed the experiments; Y.Z. fabricated the device and performed the experiments with the help of J.Li, J.Liu, Z.J.W., and Y.W.; B.X. performed calculation with the help of J.Liu and X.L.; Y.Y. performed STEM with the help of Z.J.W. and Y.H.; Y.Z., B.X., Y.Y., Y.W., J.Li, J.Liu and Z.J.W. analyzed the data; Y.D., J.H., P.D., J.W. and X.Z. participated in the correlated discussion; Y.Z., B.X., Y.Y., J.Li, J.Liu, and Z.J.W. wrote the manuscript and Supplementary Information using contributions from all authors.

## Competing interests
