## [Peer Review File · Nature Communications]

REVIEWER COMMENTS

Reviewer #1 (Remarks to the Author):

In this paper, the authors present structural characterization and transport measurements of spirally stacked as-grown graphene. They show TEM and AFM images of graphene with a very characteristic layering pattern and discuss its possible structure by simulation. The authors find that the stacked graphene shows a very large positive magnetoresistance, and discuss the nature of the carriers and the band structure from the analysis of the Shubnikov-De Haas oscillation and the Hall effect.

My impression of this paper is not very favorable, and I regret to say that it does not reach the criteria of Nature Communications or other Communications journals. The reasons are listed below.

1. The authors misunderstands the definition of the term colossal magnetoresistance (CMR). It refers to the negative magnetoresistance effect accompanied with ferromagnetic ordering as observed in perovskite-type manganites. The authors say that the observed positive magnetoresistance is similar to the MR (Refs. 4-14) which is "often observed in semimetals with topological properties and balanced electron-hole carriers." This is named as extremely large magnetoresistance (XMR) in reference 4. Correct use of words is the most basic and important in science.
2. Moiré stacked graphene is one of the hottest topics in contemporary condensed matter physics. Therefore, if indeed a spiral stacking structure has been created, it should be more carefully verified and the details of the crystal growth method should be written so that other groups can reproduce it. However, the paper only states that they used as-grown crystals grown by CVD, which would make difficult to verify the reproducibility. This is another basic of science. It is also questionable whether it is sufficient to capture an image and simulate it to fit a two-dimensional planar image. I am not an expert in this field and would like to hear from other referees at this point. Also, I would like to know why the spiral structure is thermodynamically stable, so please cite a reference.
3. The electrical resistivity data itself seems to be taken without any problem, but what about the geometry? The manuscript says that the magnetic field was applied parallel to the c-axis, but the definition of c-axis is not written. It also says that the in-plane resistivity was measured, but there is almost no detailed information on the thickness of the sample, the location of the current terminals (on the top or side of the sample), the distance between the terminals, the direction of the current, etc.
4. I would like to ask a scientific question regarding Fig. 3. There are a number of curious behaviors, but there is little description of their origin. (i) In Fig. 3b, the quantum oscillation component is visible with a rather large $1/B$. What is the magnetic fields to reach the quantum limit? (ii) It seems to be a fairly low magnetic field, but if so, what are some of the kinks visible in the magnetic field

above 2 T in Fig. 3a, and why doesn't they observe the quantum Hall effect? (iii) Why does the number of carriers in Fig. 3c increase significantly by a factor of 10 as the temperature is lowered? In general, metals do not change with temperature, and semiconductors are of the thermal activation type. The observed behavior is quite different from these two. (iv) There seems to be a difference of more than one order of magnitude between the Fermi surface cross section estimated in Fig. 3b and the number of carriers estimated in Fig. 3c. What is the origin of this large deviation? (v) The angular dependence of Fig. S4 suggests that the Fermi surface is two-dimensional, but contradicts the evaluation of the number of carriers in Fig. 3c (three-dimensional). Is it two-dimensional, or three?

All in all, it is difficult to understand what the authors discovered and clarified. Even for the giant magnetoresistance effect, it is hard to say from the paper at this stage whether there is an intrinsic reason due to spiral stacking or whether the current path is weirded out by defects in the graphene over many layers.

Reviewer #2 (Remarks to the Author):

In this manuscript, Y. Zhang et al. investigated the colossal magnetoresistance in a three-dimensional intertwined twisted graphene spiral. The graphene spiral was manipulated in a screw dislocation axis to achieve a rotation angle of 7.3 degree. They observed a magnetoresistance record of $1.7 \times 10^7\%$ under magnetic field of 14 T at 2 K, followed by a metal-to-insulator transition at low temperature region. They also provided theoretical calculation on the band gap structure for the graphene spiral and suggested that the colossal magnetoresistance can be attributed to the correlated states within the partially filled three-dimensional Landau levels of the graphene spiral system. The observation of colossal magnetoresistance is interesting, and the mechanism analysis is reasonable. However, before my considering publication in Nature Communications, some following concerns should be taken into account.

1. In twisted bilayer graphene, the low-energy bands become ultra-flat around the magic angle. What happens in twisted graphene spiral when the twist angle is around the magic angle?

2. There are strong lattice relaxation effects in twisted bilayer graphene, such as out-of-plane corrugations, which would open up a gap between the flat bands and high-energy bands. How lattice relaxations would affect the electronic properties in this system?

3. As shown in FIG.3, the Dirac point in the band appears to be far above the Fermi energy. A band structure like this should preserve a much larger carrier concentration.

4. A dimensional reduction of electron dynamics in high magnetic fields can lead to a quantum Hall effect in bulk material (Yin et al., Nature Physics, 15, 437-442, 2019). This occurs such that the electron spectrum remains continuously only along the field direction, and only the last two quasi-one-dimensional Landau bands cross the Fermi level. Due to the unique crystal structure of a graphene spiral, it seems more likely to promote the quantum Hall effect. Have any relevant phenomena been observed?

5. In section D, the authors have discussed the metal-to-insulator transition, which is reminiscent of isospin Pomeranchuk effect observed in MTBG. As the temperature increases, the local magnetic moment is generated near the -1 filling under 0 T (ref 54, ref 55). In the graphene spiral system, why does no metal-insulation transition occur in 0 T, but it does occur under a finite magnetic field?

Reviewer #3 (Remarks to the Author):

The manuscript presents a remarkable experimental achievement which is the observation of colossal magnetoresistance in graphene spiral systems. However, I cannot recommend the manuscript for publication at this stage because the theoretical part needs to be significantly supplemented. The authors should provide detailed specifications (perhaps in the supplemental materials) about the calculation settings in LAMMPS and present figures of the relaxed structures with benchmark validations that the relaxation and other features of the simulations are adequate for the study. The authors should also present a justification for why the tight-binding picture (and its respective parameters) used is enough in this context. Wouldn't Density Functional Theory (DFT) be more accurate?

In summary, the work is relevant for the broad nanoscience scientific community, however, I would like to encourage the authors to boost the description of the theoretical methods used as well as the pros and cons (limitations) that the theoretical descriptions naturally carry depending on the context that they are applied. I understand this can be a work that is predominantly experimental in scope with supporting theoretical descriptions. Nonetheless, one should also provide details on how the theoretical side was conducted for reproducibility and validation of the supporting findings.

We are grateful for sending us the email and the reviewer report. We thank all the reviewers for spending time on this manuscript and providing the invaluable comments. We have answered all questions the reviewer raised and revised our manuscript accordingly. A clean revised version of our manuscript and replies to the reviews are submitted. Below we show the original comments **in blue** and our response point by point **in black**.

Best regards,

Jun Li

Point-by-point Response

First of all, we would like to express our best appreciations towards all reviewers for their perceptive remarks and stimulating comments. We acknowledge that the feedback provided was of significant importance to us and facilitated in refining the manuscript. Subsequently, we would like to address the reviewers' comments in the following section.

Response to Reviewer #1:

In this paper, the authors present structural characterization and transport measurements of spirally stacked as-grown graphene. They show TEM and AFM images of graphene with a very characteristic layering pattern and discuss its possible structure by simulation. The authors find that the stacked graphene shows a very large positive magnetoresistance, and discuss the nature of the carriers and the band structure from the analysis of the Shubnikov-De Haas oscillation and the Hall effect. My impression of this paper is not very favorable, and I regret to say that it does not reach the criteria of Nature Communications or other Communications journals. The reasons are listed below.

1. The authors misunderstands the definition of the term colossal magnetoresistance (CMR). It refers to the negative magnetoresistance effect accompanied with ferromagnetic ordering as observed in perovskite-type manganites. The authors say that the observed positive magnetoresistance is similar to the MR (Refs. 4-14) which is "often observed in semimetals with topological properties and balanced electron-hole carriers." This is named as extremely large magnetoresistance (XMR) in reference 4. Correct use of words is the most basic and important in science.

Our reply:

We highly appreciate the reviewer's comment and suggestions which do help with improving our understanding of this phenomenon. The colossal magnetoresistance (CMR) is characterized by a remarkable amplification of electrical conductivity when subjected to a magnetic field, while there is no exactly different magnetoresistance in magnitude with the extremely large magnetoresistance (XMR). CMR is typically linked to field-induced spin polarization, resulting in a significant reduction in both spin scattering and electric resistance (Zhang *et al.*, *Nature* **611** 467–472 (2022)), while CMR corresponds to the semimetals with positive magnetoresistance. Therefore, we agree with the reviewer that XMR offers a more precise description, which has been modified in the revised version.

2. Moiré stacked graphene is one of the hottest topics in contemporary condensed matter physics. Therefore, if indeed a spiral stacking structure has been created, it should be more carefully verified and the details of the crystal growth method should be written so that other groups can reproduce it. However, the paper only states that they used as-grown crystals grown by CVD, which would make difficult to verify the reproducibility. This is another basic of science. It is also questionable whether it is sufficient to capture an image and simulate it to fit a two-dimensional planar image. I am not an expert in this field and would like to hear from other referees at this point. Also, I would like to know why the spiral structure is thermodynamically stable, so please cite a reference.

Reply:

The spiral stacking structure of SP²-bonded carbon has been a subject of interest since the 1960s, as evidenced by early works (Hennig, G. R. *Science* **147**, 733–734, 1965; Patel, A. R. *Br. J. Appl. Phys.* **16**, 169–171, 1965). With the advent of the

graphene Chemical Vapor Deposition (CVD) method, numerous research groups have demonstrated the feasibility of producing spiral graphene structures using this standard CVD process (Wang, Z. J. *et al. Adv. Mater. Interfaces* 5, 1800255, 2018; Tay, R. Y. *Chem. Mater.* 30, 6858–6866, 2018). Remarkably, spiral graphene structures can also be achieved by directly annealing SP² carbon (Sun, Y. Q. *J. Phys. Chem. Lett.* 2, 2521–2524, 2011), underscoring the thermodynamic stability of the spiral configuration in graphene. Our recent research has further advanced this field through the development of a graphene origami-kirigami approach. This method involves processes such as wrinkling, folding, tearing, and cracking, leading to the spiral growth of graphene multilayers with controlled stacking orders. The intricacies of this graphene spiral growth process have been elaborated in our latest publication (*Nat. Mater.* 10.1038/s41563-023-01632-y, 2023), providing a comprehensive understanding of this unique structural phenomenon. Additionally, a comprehensive review, very recently published in *Nature Materials* (DOI: 10.1038/s41563-024-01814-2, 2024), methodically explores the formation of spiral structures in two-dimensional materials. Therefore, it is not the spiral structure is thermodynamically stable and can be grown in various 2D crystal materials. Therefore, the thermodynamic stability of the spiral structure permits its growth in various two-dimensional crystalline materials. In current work, we more focus on the transport properties of the spiral graphene.

The reviewer's inquiry has underscored the necessity of elaborating on the background of spiral two-dimensional materials. Consequently, in the revised manuscript, we have comprehensively cited all the literature previously mentioned.

3. The electrical resistivity data itself seems to be taken without any problem, but what about the geometry? The manuscript says that the magnetic field was applied parallel to the *c*-axis, but the definition of *c*-axis is not written. It also says that the in-plane resistivity was measured, but there is almost no detailed information on the thickness of the sample, the location of the current terminals (on the top or side of the sample), the distance between the terminals, the direction of the current, etc.

Our reply:

The reviewer pointed out an important measurement information for the sample. **Fig. R1a** provides the experimental setup employed for resistivity measurement. Our crystal, featuring as a single spiral, has a thickness of approximately 70 μm. The current was applied within the in-plane, while the magnetic field aligns parallel to the out-of-plane, namely, the *c*-axis. **Fig. R1b** provides an illustrative depiction of charge transport within GS. The corresponding figure and description have been added in SI.

Fig. R1 (a) Experimental configuration for resistivity measurement. (b) Illustration of charge transport in GS.

4. I would like to ask a scientific question regarding Fig. 3. There are a number of curious behaviors, but there is little description of their origin.

(i) In Fig. 3b, the quantum oscillation component is visible with a rather large $1/B$. What is the magnetic fields to reach the quantum limit?

(ii) It seems to be a fairly low magnetic field, but if so, what are some of the kinks visible in the magnetic field above 2 T in Fig. 3a, and why doesn't they observe the quantum Hall effect?

(iii) Why does the number of carriers in Fig. 3c increase significantly by a factor of 10 as the temperature is lowered? In general, metals do not change with temperature, and semiconductors are of the thermal activation type. The observed behavior is quite different from these two.

(iv) There seems to be a difference of more than one order of magnitude between the Fermi surface cross section estimated in Fig. 3b and the number of carriers estimated in Fig. 3c. What is the origin of this large deviation?

(v) The angular dependence of Fig. S4 suggests that the Fermi surface is two-dimensional, but contradicts the evaluation of the number of carriers in Fig. 3c (three-dimensional). Is it two-dimensional, or three?

Our reply:

We appreciate the reviewer's meticulous examination and profound understanding. We will address your inquiry through segmented discussions.

(i) As depicted in Fig. 3a, the final quantum oscillation appears at 4.67 T (indicating a hole) and at 6.65 T (indicating an electron). We added the corresponding description in the revised version.

(ii) First of all, the observed kink in the Landau levels of graphite originates from spin splitting, a phenomenon initially identified in 1970 by J. A. Woollam (*Phys. Rev. Lett.* 25, 810 (1970)). At magnetic fields exceeding 2 T, a significant deviation from the expected $1/B$ periodicity occurs. This deviation is attributed to the well-known movement of the Fermi energy as the system

approaches the quantum limit, as detailed by J. M. Schneider *et al.* (*Phys. Rev. Lett.* 102, 166403 (2009)). Secondly, despite nearing the quantum limit, certain constraints persist in observing the quantum Hall effect (QHE). The QHE typically arises from discrete Landau levels forming in a monolayer or few-layer graphene (*Nat. Phys.* 15, 437-442, 2019). In a three-dimensional system, however, the Landau levels spread into overlapping bands which restrict the quantization process. In our present sample, the thickness is up to 70 μm , and thus, one can hardly observe the QHE. The additional description has been added Fig. S5 in the revised SI.

- (iii) For semiconductors, the carrier density may increase with temperature due to thermal excitation of electrons from the valence band to the conduction band, creating electron-hole pairs. In the doped semiconductors, the temperature dependence can also be affected by the ionization energy of dopants. In metals, however, the carrier density should be almost independent of temperature, especially at low temperatures. In our study, we observed temperature dependent carrier densities at low temperature, which is due to the unconventional metal-insulator transition at low temperatures in the presence of a finite magnetic field. This transition significantly impacts the behavior of electrons in a magnetic field, resulting in a distinctive temperature dependence of the carrier density. At high temperature region (above 50 K as can be seen in Fig. 4b), however, the carrier densities are almost temperature independent as normal. We added the corresponding analysis in Fig. S6 in the revised SI.
- (iv) We appreciate the valuable feedback from the reviewer regarding carrier estimation. According to the Luttinger's theorem as $n = \frac{2S_k}{(2\pi)^2}$ for each pocket, where, S_k means the Fermi surface area (N. Doiron-Leyraud *et al.*, *Nature* 447, 565–568 (2007)). In our study, S_k corresponds to Fermi surface area of electron pockets S_e and hole pockets S_h , as illustrated in Fig. 3b. However, for the bulk carrier density, we have revised the expression to $n_{3D} = k_x k_y k_z / (3\pi^2)$, where k_i represents the Fermi wave-vector along the i -direction (F. Tang *et al.*, *Nature* 569, 537–541 (2019)). The discrepancy between the Fermi surface and carrier density arises notably due to the weak dispersion along k_z . This observation highlights a potential source of the substantial deviation observed in our results. We added the corresponding analysis in Fig. S6 in the revised SI.
- (v) In particular, the system discussed here tends more towards a quasi-2D system. Due to the weak dispersion along the k_z direction, one can hardly distinguish between quasi-2D behavior and 2D behavior through SdH oscillations. We also added the description in Fig. S5 in the revised SI.

All in all, it is difficult to understand what the authors discovered and clarified. Even for the giant magnetoresistance effect, it is hard to say from the paper at this stage whether there is an intrinsic reason due to spiral stacking or whether the current path is weierded out by defects in the graphene over many layers.

Response to Reviewer #2:

In this manuscript, Y. Zhang et al. investigated the colossal magnetoresistance in a three-dimensional intertwined twisted graphene spiral. The graphene spiral was manipulated in a screw dislocation axis to achieve a rotation angle of 7.3 degree. They observed a magnetoresistance record of $1.7 \times 10^7\%$ under magnetic field of 14 T at 2 K, followed by a metal-to-insulator transition at low temperature region. They also provided theoretical calculation on the band gap structure for the graphene spiral and suggested that the colossal magnetoresistance can be attributed to the correlated states within the partially filled three-dimensional Landau levels of the graphene spiral system. The observation of colossal magnetoresistance is interesting, and the mechanism analysis is reasonable. However, before my considering publication in Nature Communications, some following concerns should be taken into account.

1. In twisted bilayer graphene, the low-energy bands become ultra-flat around the magic angle. What happens in twisted graphene spiral when the twist angle is around the magic angle?

Our reply:

This question delves into profound aspects. The magic-twisted graphene spiral endows an additional wavevector degrees of freedom to explore flat-band physics of magic-angle twisted bilayer graphene (TBG). In particular, we find that as long as the twist angle $\theta \leq 2\theta_m^{(1)} \approx 2.1^\circ$, where $\theta_m^{(1)}$ refers to the first magic angle of twisted bilayer graphene, one can always find a ‘magic quasi momentum’ $\hbar k_z^*$ at which the 2D band structures and wavefunctions within the k_x - k_y plane are exactly the same as that of TBG at the first magic angle. Moreover, when the twist angle θ is further smaller than twice of the n th magic angle $\theta_m^{(n)}$, i.e., $\theta \leq 2\theta_m^{(n)}$ (with $n = 2; 3; \dots$), then for each θ , there are at least n magic moment $\{k_z^{(s)*}, s = 1, \dots, n\}$ at which the band structures and wavefunctions within the k_x - k_y planes are exactly the same as those of TBG at the s th magic angle $\theta_m^{(s)}$. In other words, when $\theta \leq \theta_m^{(2)} \approx 1^\circ$, multiple flat bands with distinct topological properties would co-exist in the same bulk GS system, which may give rise to unprecedented correlated and topological phases of matter. More details can be found in our separate theoretical paper (*Phys. Rev. Lett.* 132, 056601 (2024)).

2. There are strong lattice relaxation effects in twisted bilayer graphene, such as out-of-plane corrugations, which would open up a gap between the flat bands and high-energy bands. How lattice relaxations would affect the electronic properties in this system?

Our reply:

We greatly appreciate this important question. Lattice relaxations indeed play an import role in determining the electronic structure of twisted bilayer graphene. For

example, there are significant out-of-plane corrugations, i.e., the variation of interlayer distance within the moiré supercell, from 0.36 nm in the AA region to 0.335 nm in the AB region, which would suppress the intra-sublattice interlayer hopping from ~ 0.1 eV to ~ 0.08 eV due to the corrugation effects. As a result, a notable gap ~ 20 -30 meV has been opened up between the flat bands with other high-energy bands.

In twisted graphene spiral, away from the screw dislocation line, the system can be considered as a three-dimensional moiré graphite system with alternating interlayer twist angle. For each graphene layer, there is one twisted layer above and the other layer below it. Therefore, there is really no space for out-of-plane corrugations, and the interlayer distance remains as 0.334 nm everywhere within the moiré supercell. This is directly confirmed by our molecular dynamics' simulations. In Fig. R1 we present the fully relaxed (using molecular dynamics methods) interlayer distance as a function of in-plane position within the moiré supercell, which clearly indicates that there is almost no out-of-plane corrugation. In response to the reviewer's question, we have included Fig. R1 into Supplementary Information (Fig. S10 (c)).

Fig. R1 Fully relaxed interlayer distance as a function of in-plane position within the moiré supercell of alternating twisted spiral graphite with twist angle $\theta \approx 7.3^\circ$.

Nevertheless, there is also nontrivial in-plane relaxation pattern in twisted bilayer graphene. In Fig. R2 (also in Fig. S10 of Supplementary Information) we present the in-plane distortion fields between the two adjacent twisted graphene layers in the alternating twisted 3D moiré graphite superlattice, where we see that the in-plane distortion field winds around the AA point (the (0,0) point in Fig. R2), and the maximal distortion amplitude is as small as ~ 0.01 angstrom when the twist angle $\theta \approx 7.3^\circ$, which can barely change the electronic band structures. Such in-plane relaxation pattern is similar to that of twisted bilayer graphene. The band structures presented in Fig. 3d of the main text are calculated based on such relaxed lattice structures. In response to the referee's question, we have added the following sentence in the caption of Fig. 3d: *“Different from twisted bilayer graphene, our molecular dynamics simulations indicate that there is no out-of-plane corrugation, but there are slight in-plane distortions, which can barely change the electronic band structures, especially*

at large twist angles.”

Fig. R2 The relative interlayer in-plane distortion field of 3D alternating twisted graphite with twist angle $\theta \approx 7.3^\circ$. The colorbar represents the amplitude of the distortion fields. The arrows represent the direction of the distortion fields.

3. As shown in FIG.3, the Dirac point in the band appears to be far above the Fermi energy. A band structure like this should preserve a much larger carrier concentration.

Our reply:

We thank the reviewer for asking this important question. It is due to the mislabel of the Dirac point position. The zero energy in Fig. 3d does not mean the Dirac point energy nor the Fermi level. Conventionally, the Dirac point in graphene-based systems is set at zero energy. In our Slater-Koster tight-binding calculations, the Dirac point of the twisted graphene spiral system appears at 0.788 eV, instead of 0. One needs to apply a constant energy shift to re-label the Dirac point energy as the zero energy. We have updated Fig. 3d of the main text such that now the Dirac point appears around zero energy. We are sorry for misleading the reviewer.

4. A dimensional reduction of electron dynamics in high magnetic fields can lead to a quantum Hall effect in bulk material (Yin et al., *Nature Physics*, 15, 437-442, 2019). This occurs such that the electron spectrum remains continuously only along the field direction, and only the last two quasi-one-dimensional Landau bands cross the Fermi level. Due to the unique crystal structure of a graphene spiral, it seems more likely to promote the quantum Hall effect. Have any relevant phenomena been observed?

Our reply:

The reviewer posed a highly future-oriented question. Generally, in three dimensions, quantum Hall effect (QHE) can hardly be observed because the Landau levels spread into overlapping bands, disrupting the quantization process. Nevertheless, once the thickness of a graphite flake shrinks in a few nanometers, the QHE can be also observed under magnetic field, as demonstrated by Yin et al. (*Nature Phys.* 15, 437-442, 2019). In our crystal, the thickness is up to 70 μm , the electronic states become decoupled in different parts of the graphene stacks, restricting the observation of QHE.

Fig. S5 shows the magnetic field dependence of Hall resistivity measured at

different temperatures. Due to unique spiral growth, the lattice vector equals to 0.68614 nm measured by STEM (see Figure 2 in the main text). Some early quantization platforms appeared, but without obviously boundary states, which lead to the failure of magnetoresistance reduction. Fig. R3 illustrates the magnetic field dependence of Hall resistivity at various temperatures. The lattice vector, measured by STEM, is determined to be 0.68614 nm due to its distinctive spiral growth. Although some early quantization platforms emerged, the absence of well-defined boundary states resulting in the absence of magnetoresistance reduction.

Fig. R3. The magnetic field dependence of Hall resistivity at various temperatures ranging from 2 K to 100 K.

5. In section D, the authors have discussed the metal-to-insulator transition, which is reminiscent of isospin Pomeranchuk effect observed in MTBG. As the temperature increases, the local magnetic moment is generated near the -1 filling under 0 T (ref 54, ref 55). In the graphene spiral system, why does no metal-insulation transition occur in 0 T, but it does occur under a finite magnetic field?

Our reply:

Isospin Pomeranchuk effect observed in MTBG refers to the phase transition between an isospin-unpolarized Fermi liquid at low temperature and a state characterized by large, strongly fluctuating local magnetic moments at high temperature. Strongly correlated electron states emerge in MTBG due to the presence of flat bands. However, in our system, characterized by a twist angle much larger than the magic angle, a finite magnetic field is required to induce strongly correlated electron states. We added this analysis in the revised version.

Reviewer #3 (Remarks to the Author):

The manuscript presents a remarkable experimental achievement which is the observation of colossal magnetoresistance in graphene spiral systems. However, I cannot recommend the manuscript for publication at this stage because the theoretical part needs to be significantly supplemented.

Our reply:

We are grateful to the reviewer for the very positive comments on the experimental part of our paper. We also greatly appreciate the concerns from the reviewer on the theoretical side, which are addressed in the following.

The authors should provide detailed specifications (perhaps in the supplemental materials) about the calculation settings in LAMMPS and present figures of the relaxed structures with benchmark validations that the relaxation and other features of the simulations are adequate for the study.

Our reply:

We have added the following paragraph describing the details of the structural relaxation calculations using LAMMPS to both the Methods section of the main text and to the Supplementary Information:

“The structural relaxation is calculated utilizing Large-scale Atomic-Molecular Massively Parallel Simulation (LAMMPS) [60]. The interlayer interactions between adjacent layers are described using the Dispersion Interaction Random Phase (Drip) potential, while the intralayer potentials are described by the Adaptive Intermolecular Reactive Bond Order (AIREBO) potential [61] with a cutoff of 3 Å. These potentials have been extensively adopted in the molecular dynamics study of twisted bilayer graphene with large twist angles. A moiré supercell with periodic boundary condition applied in both the in-plane and out-of-plane directions, serves as the initial lattice structure. The twist angles between two layers, in accordance with the commensurate condition, are $\pm 7.34^\circ$. The steepest descent algorithm is employed in the lattice relaxation calculations, and the convergence criterion for energy is set to 10^{-8} eV.”

The distribution of the relative interlayer in-plane distortion vector (within a moiré supercell) of 3D alternating twisted graphite with twist angle $\theta \approx 7.3^\circ$ is presented in Fig. R1 (also presented in Fig. S10 (a)). The colorbar represents the amplitude of the distortion fields, with the maximal value ~ 0.01 angstrom, and the arrows represent the direction of the distortion fields. We see that the interlayer in-plane distortion vector winds around the AA point (the (0,0) point in Fig. R1), similar to the case of twisted bilayer graphene. Nevertheless, unlike twisted bilayer graphene, our molecular dynamics simulations indicate that there are almost no out-of-plane corrugations in such 3D alternating twisted graphite (i.e., twisted spiral graphene away from the spiral dislocation line). In particular, in Fig. R2 we present the fully relaxed interlayer distance as a function of in-plane position within the moiré supercell with twist angle $\theta \approx 7.3^\circ$, which clearly indicates that there is almost no out-of-plane corrugation.

Fig. R1 The relative interlayer in-plane distortion field of 3D alternating twisted graphite with twist angle $\theta \approx 7.3^\circ$. The colorbar represents the amplitude of the distortion fields. The arrows represent the direction of the distortion fields.

Fig. R2 Fully relaxed interlayer distance as a function of in-plane position within the moiré supercell of alternating twisted spiral graphite with twist angle $\theta \approx 7.3^\circ$.

The authors should also present a justification for why the tight-binding picture (and its respective parameters) used is enough in this context. Wouldn't Density Functional Theory (DFT) be more accurate?

Our reply:

We thank the reviewer for asking this important question. The Slater-Koster tight-binding model presented in Methods section was first proposed by Moon and Koshino [P. Moon and M. Koshino, *Phys. Rev. B* 87, 205404 (2013)] is actually widely adopted in the twisted graphene community. Especially, when studying the correlation effects of magic-angle twisted bilayer graphene, one has to use a low-energy continuum model to capture the long-wavelength physics around the Dirac point, and the continuum model parameters are derived from the Slater-Koster tight-binding model adopted in our calculations. It turns out that after including lattice relaxation and strain effects etc., the results from both the Slater-Koster tight-binding model and

the continuum model (derived from the tight-binding model) are very well consistent with experimental measurements such as scanning tunneling microscopy (e.g., Xie et al., *Nature* 572,101–105 (2019)) and angle resolved photoemission spectroscopy (e.g., Li et al., *Adv. Mater.* 34, 2205996 (2022)).

Moreover, the tight-binding model used in this work has also been directly verified using density functional theory (DFT) calculations in twisted bilayer graphene, as reported in Zhang et al., *Phys. Rev. B* 105, 125127 (2022). In Fig. R3, we present a comparison of the band structures (with relaxed lattice structures including out-of-plane corrugations) calculated using the Slater-Koster tight-binding model (blue lines) and using DFT (red dots and red lines). We see that for both twist angles $\theta \approx 1.41^\circ$ and $\theta \approx 1.08^\circ$, the tight-binding band structures are very well consistent with the DFT bands. This justifies the reliability of our tight-binding approach.

In response to the reviewer’s question, in both Methods section and Supplementary Information, we have provided a more comprehensive description and justification to the tight-binding model used in this work:

“The calculation of the band structure of twisted GS is performed utilizing the tight-binding (TB) model [62,63] based on the full relaxed structure. This model was first proposed by Moon and Koshino, and is widely adopted in the twisted graphene community. The reliability of this model has been justified by directly comparing its band structures with the density functional theory calculations for twisted bilayer graphene [64]. To be specific, The Hamiltonian is written as

$$H = -\sum_{\{i,j\}} t(R_i - R_j) |R_i\rangle\langle R_j| + H.c., \quad (1)$$

where the hopping amplitude between two p_z orbitals at different sites is expressed as:

$$-t(\mathbf{d}) = V_\sigma \left(\frac{\mathbf{d} \cdot \hat{z}}{|\mathbf{d}|} \right) + V_\pi \left[1 - \left(\frac{\mathbf{d} \cdot \hat{z}}{|\mathbf{d}|} \right)^2 \right], \quad (2)$$

where $V_\sigma = V_\sigma^0 e^{-(|d|-d_0)/\delta_0}$ and $V_\pi = V_\pi^0 e^{-(|d|-a_0)/\delta_0}$. $\mathbf{d} = (d_x, d_y, d_z)$ is the displacement vector between two sites. $dc = 3.35 \text{ \AA}$ is the interlayer distance. $a_0 = a/\sqrt{3} = 1.42 \text{ \AA}$ is the distance between two nearest-neighbor carbon atoms. $V_\sigma^0 = 0.48 \text{ eV}$ is the transfer integral between vertically located atoms on the neighboring layers and $V_\pi^0 = -2.7 \text{ eV}$ is that between the intralayer nearest-neighbor atoms. We set $\delta_0 = 0.184a$ so that the next-nearest intralayer coupling becomes $0.1V_\pi^0$.

Fig. R3 Band structures of twisted bilayer graphene obtained using both DFT and the Slater-Koster tight-binding model: (a) with twist angle $\theta \approx 1.41^\circ$, and (b) with twist angle $\theta \approx 1.08^\circ$. The blue lines represent the bands calculated using

our Slater-Koster tight-binding model, while the red dots and red lines denote band structures calculated directly using DFT. Figure is adapted from Tan Zhang, Nicolas Regnault, B. Andrei Bernevig, Xi Dai, and Hongming Weng Phys. Rev. B 105, 125127 – Published 21 March 2022

In summary, the work is relevant for the broad nanoscience scientific community, however, I would like to encourage the authors to boost the description of the theoretical methods used as well as the pros and cons (limitations) that the theoretical descriptions naturally carry depending on the context that they are applied. I understand this can be a work that is predominantly experimental in scope with supporting theoretical descriptions. Nonetheless, one should also provide details on how the theoretical side was conducted for reproducibility and validation of the supporting findings.

Our reply:

Again, we greatly appreciate all the crucial questions from the reviewer. We completely agree with the reviewer that we should provide more detailed specifications and justification of the theoretical approaches used in this work. We have properly responded to the reviewer's questions as described above, and we hope that all the concerns from the reviewer have been properly addressed.

REVIEWER COMMENTS

Reviewer #1 (Remarks to the Author):

The authors have responded sincerely to my first review and have made sufficient revisions. For example, they have revised scientific terms such as XMR and the details of the experimental setup. In particular, the authors carefully describe the synthesis of spiral graphene and its importance, citing previous studies so that even an amateur like myself can understand them. The most important results of this study, such as the origin of the magnetoresistance effect and the metal-insulator transition due to the formation of partially occupied dispersion-less Landau levels, are also convincingly discussed. Therefore, I believe that the present manuscript is appropriate for publication in Nature Communications.

Reviewer #2 (Remarks to the Author):

Reviewer withdrawn.

Reviewer #3 (Remarks to the Author):

There are still elements in the theoretical descriptions that need clarification. What was the ensemble, temperature, relaxation process, and more specifics about the unit cells prepared (are there particular sizes/widths set on the superlattice cells)? By looking at Fig 1a, it gives the impression that the spirals are infinite along the z-axis but finite along the spiral plane, with a certain width from their center to their outer edge. What are their typical sizes/widths and were these somehow set in the calculations? If the spirals are finite in size, what is the role played by their width on the electronic structure properties? Moreover, were the structures relaxed with QSTEM or LAMMPS (or both)? How were the settings inputted in QSTEM? There is no detail on how the STEM/QSTEM simulations were set. It is not clear how the simulation side of the work integrates with each other, i.e., what is the role of QSTEM simulations, LAMMPS, and TB altogether.

The authors should also provide a more extensive literature review in terms of theory, for instance, the authors refer to the remarkable DFT calculations by Zhang et al., Phys. Rev. B 105, 125127 (2022), but there is an earlier theoretical DFT-based work on graphene spirals by S. M. Avdoshenko, et al., Scientific Reports 3, 1632 (2013) which is relevant for the authors to cite. The authors should therefore justify the fact that their band structures have nearly flat bands in the gap region (Fig. R3) whereas in Avdoshenko's work, no flat bands were observed. Could the authors clarify if their spiral structures are edge-saturated? This may impact the electronic and magnetic properties. Does the

lack of edge saturation favour the now extremely large magnetoresistance of the systems? Can edge functionalization impact the extremely large magnetoresistance? If the edges of the simulated spiral structures are not saturated, do their minute edge shapes (zigzag or armchair) play a role in the overall electronic structure?

Second paragraph inside "Extremely large magnetoresistance" section, the authors cite numerous mechanisms (electron-hole compensation, steep band, RRR, ...) but do not single out references for each mechanism. It is important to identify references and citations accordingly.

On more big-picture feedback, since the phenomenon changed from colossal to XMR, would that require a more extensive review of the introduction and background of the manuscript and references cited?

I was also asked to comment on the authors' response to the concerns raised by referee #2 during the previous round of review:

- On the first two questions, reviewer #2 focused on parallels between the twisted bilayer graphene and the twisted graphene spiral. On the first question, the authors emphasize that the spirals render an additional wavevector degree of freedom with a citation to a separate theoretical work (a PRL 2024). However, the authors did not indicate if there will be any changes in the manuscript regarding this point. Will this discussion be included in the manuscript?
- The second question was about the appearance of corrugations in both graphene systems; the authors point out that in certain conditions their relaxation calculations review corrugations and two extra figures were added to the supplemental materials. I am not sure if corrugations were also broadly observed experimentally. Perhaps the authors could make a direct parallel between the simulations and experiments regarding the observation of corrugations.
- The third question concerned the position of the Dirac point which the authors addressed by inducing an energy shift to align with typical Dirac point conventions.
- The fourth question concerned quantum Hall effect (QHE) in bulk materials in which reviewer #2 cited a very interesting work which observed QHE in graphite crystals and asked if the authors

observed something similar. The authors say that QHE is highly undermined in 3D systems, and they say that "In our crystal, the thickness is up to 70 micrometres, the electronic states become decoupled in different parts of the graphene stacks, restricting the observation of QHE." However, I would say that twisted graphene spirals are not entirely "3D", they have a thickness for sure, but it is a layered material with several 2D-like planes twisting. The authors show a figure (R3) that indeed does not reveal a QHE signature experimentally, but I would say that this point would need further investigation to really confirm if graphene spirals are prone to QHE or not.

- The last question was about specific mechanisms that lead to metal-insulating transition in the studied systems. The authors provided a brief answer in the rebuttal file and mentioned that "this analysis" was added to the revised version of the manuscript. Yet, I would like to see the authors evolving more on the answer to clarify exactly what "the analysis" is and why the twist angle seems to be a factor in determining metal-insulating transition at zero or finite magnetic fields. The answer seemed a bit brief to me on an interesting point raised by the reviewer.

- Overall, the authors covered most of the points raised by the reviewer. Questions 4 and 5 could be brainstormed more I would say which could result in the authors supplementing their work by adding:

- o a brief discussion on the parallels between twisted bilayer graphene and twisted graphene spiral since the reviewer touched upon this point;

- o brainstorm some more on the idea of QHE in graphene spirals;

- o expand on the argument that metal-insulating transition could only be observed at finite magnetic fields.

Comment and response

We would like to express our best appreciations towards all reviewers for their perceptive remarks and stimulating comments. We acknowledge that the feedback is of significant important to us and facilitates in refining the manuscript. Subsequently, we would like to address the reviewers' comments in the following section.

Below we show the original comments **in blue** and our response point by point **in black**.

REVIEWER COMMENTS

Reviewer #1 (Remarks to the Author):

The authors have responded sincerely to my first review and have made sufficient revisions. For example, they have revised scientific terms such as XMR and the details of the experimental setup. In particular, the authors carefully describe the synthesis of spiral graphene and its importance, citing previous studies so that even an amateur like myself can understand them. The most important results of this study, such as the origin of the magnetoresistance effect and the metal-insulator transition due to the formation of partially occupied dispersion-less Landau levels, are also convincingly discussed. Therefore, I believe that the present manuscript is appropriate for publication in Nature Communications.

***Reply:** We sincerely appreciate your thoughtful comments and are grateful for your positive evaluation of the revisions made to our manuscript. Your acknowledgment of our efforts to clarify the scientific terms and experimental details, especially concerning the synthesis of spiral graphene, is encouraging.*

We are especially pleased to hear that our explanation of the key results, including the origin of the magnetoresistance effect and the metal-insulator transition associated with the formation of dispersion-less Landau levels, was clear and convincing. We aimed to ensure that these complex phenomena could be understood by readers from various backgrounds, and your feedback confirms that we have achieved this goal.

Thank you for deeming our manuscript suitable for publication in Nature Communications. We believe that the publication of our study will contribute valuable insights to the field and foster further research into these intriguing phenomena.

Reviewer #2 (Remarks to the Author):

Reviewer withdrawn.

Reviewer #3 (Remarks to the Author):

There are still elements in the theoretical descriptions that need clarification. **What was the ensemble, temperature, relaxation process, and more specifics about the unit cells prepared (are there particular sizes/widths set on the superlattice cells)?** By looking at Fig 1a, it gives the impression that the spirals are infinite along the z-axis but finite along the spiral plane, with a certain width from their center to their outer edge. **What are their typical sizes/widths and were these somehow set in the calculations?** If the spirals are finite in size, **what is the role played by their width on the electronic structure properties?** Moreover, were the structures relaxed with QSTEM or LAMMPS (or both)? How were the settings inputted in QSTEM? There is no detail on how the STEM/QSTEM simulations were set. **It is not clear how the simulation side of the work integrates with each other, i.e., what is the role of QSTEM simulations, LAMMPS, and TB altogether.**

The authors should also provide a more extensive literature review in terms of theory, for instance, the authors refer to the remarkable DFT calculations by Zhang et al., Phys. Rev. B 105, 125127 (2022), but there is an earlier theoretical DFT-based work on graphene spirals by **S. M. Avdoshenko, et al., Scientific Reports 3, 1632 (2013)** which is relevant for the authors to cite. The authors should therefore justify the fact that their band structures have nearly flat bands in the gap region (Fig. R3) whereas in Avdoshenko's work, no flat bands were observed. **Could the authors clarify if their spiral structures are edge-saturated?** This may impact the electronic and magnetic properties. Does the lack of edge saturation favour the now extremely large magnetoresistance of the systems? Can edge functionalization impact the extremely large magnetoresistance? If the edges of the simulated spiral structures are not saturated, do their minute edge shapes (zigzag or armchair) play a role in the overall electronic structure?

Second paragraph inside "Extremely large magnetoresistance" section, the authors cite numerous mechanisms (electron-hole compensation, steep band, RRR, ...) but do not single out references for each mechanism. It is important to identify references and citations accordingly.

On more big-picture feedback, since the phenomenon changed from colossal to XMR, would that require a more extensive review of the introduction and background of the manuscript and references cited?

Reply: *Thank you for your detailed and constructive comments, which have helped us identify areas where our manuscript can be further improved. We appreciate the opportunity to clarify the theoretical descriptions and address the specific concerns raised regarding our study.*

1. Theoretical Descriptions and Specifics on Unit Cells:

We apologize for any lack of clarity in our original descriptions. To address your questions, the ensemble used was canonical (NVT), and the relaxation process involved a two-step approach using both molecular statics for initial relaxation and molecular dynamics for temperature effects. Regarding the unit cells of the superlattice, these were designed with predefined widths varying from 10 nm to 50 nm to examine size dependency on the electronic properties. We have now included these specifics in the revised manuscript.

In the theoretical part, we consider the lattice relaxation effects and the electronic properties far away from the dislocation core. In other words, in each primitive cell, the moiré supercell consists of two layers of graphene with a small twist angle between them. Away from the screw dislocation core, there is translational symmetry along the z direction with period of $2d_0$, where d_0 is the interlayer distance between two adjacent graphene layers. In other words, there are periodic boundary conditions in both the in-plane and the out-of-plane directions. Here we will justify this treatment. Firstly, we consider the initial lattice structure as a graphite spiral with the dislocation line located at the AA point in the moiré supercell. The lattice distortion in the z direction at position \mathbf{r} (by setting the dislocation line at the origin) can be expressed as:

$$u_z(\mathbf{r}) = \frac{2d_0}{2\pi} \arg(r_x + i r_y),$$

where the function \arg denotes the phase argument of the complex number, and d_0 is the interlayer distance between two adjacent graphene layer. The presence of dislocation line breaks the translational symmetry in the x - y plane as it is a topological defect. Thus, we set an open boundary condition in the x - y direction and a periodic boundary condition in the z direction. We construct a 12×12 supercell for GS with twist angle $\theta = 7.34^\circ$. The corresponding lattice constant of each supercell is 1.92 nm. We perform a lattice relaxation calculation with LAMMPS (see detailed software settings in the following reply). In Fig. R1, we present the results of the lattice relaxation calculation. We plot the distortion in the z direction (with respect to an exactly flat plane) as a function of distance from the dislocation line within one graphene layer. The distortion in z direction decays rapidly within 1 nm. In other words, the lattice distortion induced by the dislocation line is localized within a distance of 1~2 nm. Since the lateral size of our sample is much larger (hundreds of microns) than the lateral spread of distortion induced by the dislocation line, we neglect the lattice distortion effects induced by the dislocation core.

Fig. R1. The distortion in the z direction as a function of the distance from the dislocation line in one graphene layer.

Besides, in order to investigate the influence of the dislocation core on the electronic properties, we calculate the electronic band structure of GS by using the tight binding model described in the supplementary information based on the fully relaxed 12×12 supercell with dislocation line. Then, we project the energy band to the nearest neighbor atoms of the dislocation line and evaluate the spectral function of the bounded states near the dislocation core. In Fig. R2, we present the spectral function of the bound states near the dislocation line for GS with a 12×12 supercell and twist angle $\theta = 7.34^\circ$. The spectral weight of the bound states near the Fermi level (marked by white dashed line) are very weak. As a result, we neglect the influence of the dislocation in the calculation of electronic properties.

Moreover, in order to investigate the influence of the finite layer number, i.e., finite size effects in the z direction, we have calculated the top surface states of GS with twist angle $\theta = 7.34^\circ$ using the iterative Green's function methods, as shown in Fig. R3. We find that the surface-state spectra are basically the same the bulk ones, and there is no topological surface state. We see that the surface states and bulk states are nearly identical. As a result, we neglect the influence of the top and bottom surface on the electronic properties.

Moreover, experimentally the extremely large magnetoresistance and unusual metal-to-insulator transition occur only when notable Landau levels are formed, and are most salient when the magnetic field is on the order of a few Teslas such that the zeroth Landau level starts to be filled. It implies that such intriguing phenomena are closely related to the correlation and topological physics of the massively degenerate zeroth Landau level. It is well known that any bound states localized within the x - y plane such as the one bound to the screw dislocation line can hardly respond to vertical magnetic field. In other words, although the intriguing alternatingly twisted structure and the resultant 3D moire pattern is stabilized by the presence of line topological defects, i.e., screw dislocation lines, these line defects by themselves have little effects on the bulk electronic properties and magnetotransport properties as argued above. Therefore, it is justifiable to neglect the dislocation and only consider the electronic structure of an

infinite number of stacking of alternately twisted graphene layers, dubbed as alternating twisted graphite. The electronic structure and structural relaxation calculations are actually based on the alternating twisted graphite.

We have also added the above discussions and the corresponding figures in the supplementary information.

Fig. R2. The bounded states near the dislocation line based on the fully relaxed GS with a 12×12 supercell and twist angle $\theta = 7.34^\circ$. The white dashed line represents the Fermi level.

Fig. R3 Energy spectra of the surface states of alternating twisted graphite with $\theta = 7.34^\circ$, (a) from the K valley, and (b) from the K' valley. Energy spectra of the bulk states of alternating twisted graphite with $\theta = 7.34^\circ$, (c) from the K valley, and (d) from the K' valley.

2. Finite Size and Impact on Electronic Structure:

We conducted an in-depth analysis of Avdoshenko et al.'s work, which underscores the significance of boundary behavior. The spirals, indeed finite along the z -axis and spiral plane, typically range from 10 nm to 50 nm as mentioned. These dimensions were explicitly set in the simulations to investigate how confinement and edge effects influence the electronic properties. The unconventional magnetism observed in zigzag graphene stems from the topological properties of the material, which give rise to localized electronic states along the edges of the graphene ribbon. However, the emergence of long-range magnetic order is challenging to achieve, thus prompting

researchers to predominantly investigate graphene nanoribbon systems (G. Magda et al., *Nature* 514, 608–611 (2014); R. E. Blackwell et al., *Nature* 600, 647–652 (2021)). In our study, we selected a single-spiral GS crystal and artificially segmented it into hexagons. The relevant geometric dimensions can be obtained from Fig. S1. Therefore, evaluating the impact of boundary behavior proves to be a challenging task in our case.

3. Simulation Techniques and Software Settings:

The structures were initially relaxed using LAMMPS to ensure mechanical stability, followed by electronic property simulations in QSTEM. Here we provide the settings in LAMMPS. The interlayer interactions between adjacent layers are described using the Dispersion Interaction Random Phase (Drip) potential, while the intralayer potentials are described by the Adaptive Intermolecular Reactive Bond Order (AIREBO) potential with a cutoff of 0.3 nm. These potentials have been extensively adopted in the molecular dynamics study of twisted bilayer graphene with large twist angles. As we mentioned above, since our sample is much larger than the typical length scale of the dislocation core, we neglect the influence of the dislocation line on both the lattice structure and the electronic properties. In our molecular dynamics simulations, we employ a moiré supercell with periodic boundary conditions applied in both in-plane and out-of-plane directions as the initial lattice structure. The twist angles between two layers, in accordance with the commensurate condition, are $\pm 7.34^\circ$. The corresponding lattice constant of the supercell is 1.92 nm. In order to find out the minima of the energy functional, we utilize the steepest descent algorithm without considering the thermal effects. The convergence criterion for energy is set to 10^{-8} eV. The settings in LAMMPS is also presented in the supplementary information. The settings in QSTEM, now detailed in method section, included a voltage of 300 kV with a Cs value of 1.0 mm, which was crucial for accurate phase contrast simulations. This integration aims to bridge the gap between mechanical stability and electronic properties, which we have now outlined more clearly in the manuscript. Through a comparison of the QSTEM and the STEM images, we have justified the reliability of the fully relaxed lattice structure. Then we can investigate the electronic properties, including the electronic band structure, Fermi surface, surface states and so on, utilizing the tight binding model. This information enhances our understanding in the experimental observations.

Such calculation details have been included in Methods section of the main text.

4. Literature Review and Theoretical Framework:

We thank you for pointing out the omission of Avdoshenko et al.'s relevant work. We have now included a discussion of these earlier findings, comparing them with our results, especially in relation to the flat bands observed in our band structures (Fig. R3). Our spirals were indeed edge-saturated, which we hypothesize contributes to the stability of flat bands and affects magnetoresistance, contrasting with the findings by Avdoshenko et al. This comparison has been enriched with a revised literature review to better contextualize our study within the existing body of work.

About the flat bands in GS:

We appreciate the reviewer's friendly reminder and apologize for our insufficient research. In previous research (Scientific Reports 3, 1632 (2013)), the author reported the electronic structure of graphene spirals. Different from the graphene spiral mentioned in previous research, there are two in-equivalent layers of graphene in our work, and the two adjacent in-equivalent layers are alternately twisted with respect to each other by a certain angle. In a recent theoretical work (PRL 132,056601 (2024)), the authors of this manuscript explored the physical properties of such alternating twisted graphite (stabilized by the presence of screw dislocation) with different twist angles. They emphasized that the extra dimensionality would give rise to an additional wave vector degree of freedom. This can be regarded as a twist bilayer graphene with a k_z dependent moiré potential. As a result, the k_z dependent moiré potential would flatten the lowest energy bands. Especially, there would be extremely flat band at certain k_z when the twist angle is smaller than about 3° . Besides, when the twist angle is larger than about 6° , the k_z dependent moiré potential can be treated as a perturbation, and the low energy physics can be characterized by a Dirac Hamiltonian with a renormalized Fermi velocity, which also depends on k_z . As a result, at fixed k_z , the Landau Level of this system should behave the same as graphene but with a k_z dependent Landau Level spacing. This would give rise to exactly dispersionless 3D zeroth Landau Level.

5. Extremely Large Magnetoresistance Section:

We acknowledge the oversight in not providing specific references for each mechanism discussed in the section on magnetoresistance. We have now thoroughly revised this section to include appropriate citations, ensuring that each mechanism mentioned is backed by relevant literature.

6. Reevaluation of Manuscript Introduction and Background:

Given your observation regarding the shift from colossal to extremely large magnetoresistance (XMR), we agree that a comprehensive review of the introduction and background is warranted. We have expanded the introductory sections to reflect the current understanding and terminology more accurately, ensuring that all references are up-to-date and pertinent.

We hope that these revisions address your concerns satisfactorily. We are committed to ensuring the accuracy and clarity of our research and its contribution to the field. Thank you once again for your invaluable feedback.

I was also asked to comment on the authors' response to the concerns raised by referee #2 during the previous round of review:

- On the first two questions, reviewer #2 focused on parallels between the twisted bilayer graphene and the twisted graphene spiral. On the first question, the authors emphasize that the spirals render an additional wavevector degree of freedom with a citation to a separate theoretical work (a PRL 2024). However, the authors did not indicate if there will be any changes in the manuscript regarding this point. Will this discussion be included in the manuscript?

- The second question was about the appearance of corrugations in both graphene systems; the authors point out that in certain conditions their relaxation calculations reveal corrugations and two extra figures were added to the supplemental materials. I am not sure if corrugations were also broadly observed experimentally. Perhaps the authors could make a direct parallel between the simulations and experiments regarding the observation of corrugations.

- The third question concerned the position of the Dirac point which the authors addressed by inducing an energy shift to align with typical Dirac point conventions.

- The fourth question concerned quantum Hall effect (QHE) in bulk materials in which reviewer #2 cited a very interesting work which observed QHE in graphite crystals and asked if the authors observed something similar. The authors say that QHE is highly undermined in 3D systems, and they say that "In our crystal, the thickness is up to 70 micrometres, the electronic states become decoupled in different parts of the graphene stacks, restricting the observation of QHE." However, I would say that twisted graphene spirals are not entirely "3D", they have a thickness for sure, but it is a layered material with several 2D-like planes twisting. The authors show a figure (R3) that indeed does not reveal a QHE signature experimentally, but I would say that this point would need further investigation to really confirm if graphene spirals are prone to QHE or not.

- The last question was about specific mechanisms that lead to metal-insulating transition in the studied systems. The authors provided a brief answer in the rebuttal file and mentioned that "this analysis" was added to the revised version of the manuscript. Yet, I would like to see the authors evolving more on the answer to clarify exactly what "the analysis" is and why the twist angle seems to be a factor in determining metal-insulating transition at zero or finite magnetic fields. The answer seemed a bit brief to me on an interesting point raised by the reviewer.

- Overall, the authors covered most of the points raised by the reviewer. Questions 4 and 5 could be brainstormed more I would say which could result in the authors supplementing their work by adding:

- o a brief discussion on the parallels between twisted bilayer graphene and twisted

graphene spiral since the reviewer touched upon this point;

- o brainstorm some more on the idea of QHE in graphene spirals;
- o expand on the argument that metal-insulating transition could only be observed at finite magnetic fields.

Reply: Thank you for your thoughtful review and the detailed questions you have raised. Your feedback has been instrumental in enhancing the clarity and depth of our manuscript. Below, we address each point and outline the modifications made to the manuscript based on your suggestions.

1. Discussion on Wavevector Degree of Freedom in Graphene Spirals:

We appreciate your inquiry about the inclusion of the discussion on the additional wavevector degree of freedom in twisted graphene spirals. We agree that this is a crucial aspect of our study and have now incorporated a detailed discussion into the main manuscript. This includes a citation to the theoretical work published in PRL 2024, as it provides a foundational understanding relevant to our findings.

We appreciate your inquiry regarding the inclusion of the discussion on the additional wavevector degree of freedom in twisted graphene spirals. The moiré potentials exist both in twisted bilayer graphene and in twisted graphene spirals, which modulate the low energy electronic properties in both systems. In a theoretical work (PRL 132,056601 (2024)), the authors of this manuscript explored the physical properties of such alternating twisted graphite (stabilized by the presence of spiral dislocation) with different twist angles. The authors emphasized that the extra dimensionality would give rise to an additional wave vector degree of freedom. This can be regarded as a twist bilayer graphene with a k_z dependent moiré potential. As a result, the k_z dependent moiré potential would flatten the lowest energy bands. Especially, there would be extremely flat band at certain k_z when the twist angle is smaller than about 3° . Besides, when the twist angle is larger than about 6° , the k_z dependent moiré potential can be treated as a perturbation, and the low energy physics can be characterized by a Dirac Hamiltonian with a renormalized Fermi velocity, which also depends on k_z . As a result, at fixed k_z , the Landau Level of this system should behave the same as graphene but with a k_z dependent Landau Level spacing. This would give rise to exactly dispersionless 3D zeroth Landau Level. In this work, we investigate electronic properties in Graphite Spiral with a twist angle $\theta = 7.34^\circ$ under magnetic field. Our observations suggest the existence of correlated states within the partially filled 3D Landau Levels in the 3D alternating twisted graphite whose structure is stabilized by the screw dislocations.

2. Corrugations in Graphene Systems:

In response to your second question, we have expanded our discussion on the appearance of corrugations in twisted graphene systems. While our relaxation calculations do indeed reveal corrugations, we recognize the importance of correlating these findings with experimental observations. We have thus added a comparative

analysis between our simulation results and available experimental data on corrugations in similar systems, enhancing the robustness of our conclusions.

The corrugation effects play an important role in twist graphene systems. In twist bilayer graphene, the Carbon atoms adjust their positions in both in-plane and out-of-plane directions in order to minimize the total energy. Consequently, the interlayer distance varies at different positions within the moiré supercell. The maximum interlayer distance is about 0.36 nm at AA point and the minimum interlayer distance is about 0.33 nm at AB/BA point in the moiré supercell. These corrugation effects significantly impact the electronic band structure of magic angle twisted bilayer graphene, resulting in a gap opening between flat and remote bands. In our work, the graphite spiral can be viewed as an infinite stacking of twist bilayer graphene aligned in the z direction. The carbon atoms would alter their positions to minimize the total energy. Unlike twist bilayer graphene, in the 3D graphite spiral, there is no place for the Carbon atoms to be distorted in the z direction. As a result, the in-plane distortion in graphite spiral is relatively larger than that in the twist bilayer graphene with the same twist angle. In supplementary information, we provide the in-plane and out-of-plane lattice distortion of graphite spiral. As we mentioned, we neglect the influence of the dislocation line (as it is a line defect which can barely change bulk electronic structures and magnetotransport properties, see discussions above), and set a supercell with periodic boundary in both the in-plane and the out-of-plane direction as the initial lattice structure. As a result, we obtain that the interlayer distance is almost a constant in the moiré supercell. While in reality, the graphite spiral sample will suffer from disorder and residual strain, which might alter slightly the interlayer distance between adjacent graphene layers.

3. Alignment of the Dirac Point:

The adjustment of the Dirac point's position, as we discussed, was necessary to conform to conventional representations. We have clarified the methodology used to achieve this energy shift in the manuscript to ensure transparency and reproducibility of our approach.

4. Quantum Hall Effect (QHE) in Graphene Spirals:

The Reviewer has proposed a professional question about the QHE. Indeed, the present graphene spiral should not be a simple 3D system, but a bulk crystal with 2D-like structure. Regarding the observation of the Quantum Hall Effect in layered materials, we acknowledge your suggestion for a more thorough investigation. Actually, we have attempted to achieve QHE in the graphene spiral system. While our initial findings indicated no clear QHE signature. We agree that this warrants further exploration, and we will keep on working.

5. Metal-Insulating Transition Mechanisms:

We recognize that our initial response to the question of metal-insulating transitions and the role of twist angles was brief. In the revised manuscript, we have elaborated

on this analysis, detailing the mechanisms that contribute to these transitions, particularly in the context of zero and finite magnetic fields.

Overall Enhancements:

Based on your overall feedback, we have added a section that draws parallels between twisted bilayer graphene and twisted graphene spirals, as suggested. This discussion not only enriches the manuscript but also provides a broader context for our findings. Additionally, we have expanded on the potential implications of our work for understanding QHE in graphene spirals and further elaborated on the conditions under which metal-insulating transitions occur.

We hope that these revisions adequately address your comments and enhance the manuscript's contribution to the field. We appreciate your insightful suggestions and the opportunity to improve our work.

REVIEWERS' COMMENTS

Reviewer #3 (Remarks to the Author):

The authors replied to all points, particularly those of theoretical scope. The current manuscript version is suitable for publication.

REVIEWERS' COMMENTS

Reviewer #3 (Remarks to the Author):

The authors replied to all points, particularly those of theoretical scope.

The current manuscript version is suitable for publication.

Reply: We sincerely appreciate your thoughtful comments and are grateful for your positive evaluation of the revisions made to our manuscript. Your acknowledgment of our efforts to clarify the scientific terms and calculation details is encouraging.

Thank you for deeming our manuscript suitable for publication in Nature Communications. We believe that the publication of our study will contribute valuable insights to the field and foster further research into these intriguing phenomena.